# Personalized Driver Gene Prediction Using Graph Convolutional Networks with Conditional Random Fields

**DOI:** 10.3390/biology13030184

**Published:** 2024-03-14

**Authors:** Pi-Jing Wei, An-Dong Zhu, Ruifen Cao, Chunhou Zheng

**Affiliations:** 1Information Materials and Intelligent Sensing Laboratory of Anhui Province, Institutes of Physical Science and Information Technology, Anhui University, 111 Jiulong Road, Hefei 230601, China; weipj@ahu.edu.cn (P.-J.W.); q21301153@stu.ahu.edu.cn (A.-D.Z.); 2School of Computer Science and Technology, Anhui University, 111 Jiulong Road, Hefei 230601, China; rfcao@ahu.edu.cn; 3School of Artificial Intelligence, Anhui University, 111 Jiulong Road, Hefei 230601, China

**Keywords:** cancer, driver genes, multi-omics features, graph convolutional neural network, conditional random field layer

## Abstract

**Simple Summary:**

Identifying cancer driver genes plays a significant role in cancer diagnosis and treatment. With the advancement of next-generation sequencing technologies, a wealth of multi-omics cancer data, including genomic, epigenomic, and transcriptomic data, are now available for cancer research. Integrating these data to effectively identify cancer driver genes causally associated with cancer is a computational challenge. Methods for identifying cancer driver genes are mainly based on population levels. Considering the trend of precision medicine and the heterogeneity of patients, it is challenging but crucial to identify cancer driver genes at the individual level. We developed a method called PDGCN (Personalized Drivers of GCN), which constructs sample–gene interaction networks by integrating multiple types of data features and using network structural features extracted from Node2vec. Then, a graphical convolutional neural network model with a conditional random field layer is used to prioritize candidate driver genes in the network. The results show that PDGCN can identify driver genes at the individual level, providing a new perspective for predicting driver genes in individual samples.

**Abstract:**

Cancer is a complex and evolutionary disease mainly driven by the accumulation of genetic variations in genes. Identifying cancer driver genes is important. However, most related studies have focused on the population level. Cancer is a disease with high heterogeneity. Thus, the discovery of driver genes at the individual level is becoming more valuable but is a great challenge. Although there have been some computational methods proposed to tackle this challenge, few can cover all patient samples well, and there is still room for performance improvement. In this study, to identify individual-level driver genes more efficiently, we propose the PDGCN method. PDGCN integrates multiple types of data features, including mutation, expression, methylation, copy number data, and system-level gene features, along with network structural features extracted using Node2vec in order to construct a sample–gene interaction network. Prediction is performed using a graphical convolutional neural network model with a conditional random field layer, which is able to better combine the network structural features with biological attribute features. Experiments on the ACC (Adrenocortical Cancer) and KICH (Kidney Chromophobe) datasets from TCGA (The Cancer Genome Atlas) demonstrated that the method performs better compared to other similar methods. It can identify not only frequently mutated driver genes, but also rare candidate driver genes and novel biomarker genes. The results of the survival and enrichment analyses of these detected genes demonstrate that the method can identify important driver genes at the individual level.

## 1. Introduction

Cancer is an evolving and complicated illness causing high morbidity and mortality rates worldwide. GLOBOCAN 2020 projects that 28.4 million new cancer cases will be diagnosed worldwide in 2040, a 47% increase from 2020 [1]. The early identification and the subsequent diagnostic treatment of cancer lesions are two of the most successful approaches to minimizing cancer mortality, but they require a deeper knowledge of the molecular underpinnings of tumor development and progression [2]. Complex genetic changes, such as Single Nucleotide Variations (SNVs), copy number variations (CNVs), insertions and deletions, and structural abnormalities, are the main causes of the complicated genesis of cancer [3]. Driver mutations are often referred to as mutations that provide tumor cells with a selective growth advantage and hasten the development of cancer [4]. The term “passenger mutations” refers to mutations that happen randomly in tumor samples but are not necessarily connected to the development of cancer [4,5]. Cancer driver genes (CDGs) are those with cancer-causing mutations [5]. Finding cancer driver genes in various tumor types is one of the main goals of cancer genomics. For the clinical diagnosis, prevention, and therapy of cancer, the identification of driver genes is essential [6].

The advancement of computer technology and sequencing techniques in recent years has resulted in a rise in the number of researchers working on cancer driver gene identification. Most computational techniques focus on discovering driver genes for a cancer type at the population level. Typical methods include mutation-based and network-based methods. Mutation-based approaches identify cancer driver genes based on the gene mutation frequency, with more frequently mutated genes being more likely to be driver genes [7]. For example, the methods of Mutsig [8] and MuSic [9] estimate the background mutation rates of each gene and identify driver mutations that significantly deviate from this rate. OncodriveCLUST [10] constructs a background model using silent mutations to identify genes with a tendency to cluster significant mutations in protein sequences. However, it is difficult to accurately estimate the background mutation rate and identify infrequently or rarely mutated genes using mutation-based methods [11]. Considering that the biological network can describe the relationships between genes and gene features, some network-based approaches have been proposed to identify cancer driver genes by assessing their roles in biological networks. For instance, DriverNet [12] constructs a bipartite graph based on mutated genes and differentially expressed genes (DEGs) in tumor samples and prioritizes mutated genes according to their degree. HotNet2 [13] incorporates gene interaction networks to identify significantly altered gene modules in a cohort. Furthermore, because the graph convolutional network (GCN) algorithm can directly deal with graph-structured data, showing outstanding performances, some cancer driver gene identification methods based on GCNs have been proposed. The EMOGI [14] method successfully identifies known driver genes by combining the features of different genes using a GCN model. The MTGCN [15] method constructs a multichannel GCN network that can combine the driver gene identification task with link prediction. It was also shown that combining biometric and network features could improve prediction accuracy. However, population-based approaches have limitations in that they cannot identify rare driver genes occurring in small cohorts or in individual patients.

Cancer is a complex disease with high heterogeneity, as different patients may be driven by different genes and have different outcomes even if they receive the same treatment. Therefore, it is necessary to investigate personalized cancer driver genes that are specific to individual patients. In recent years, many researchers have proposed some driver gene identification methods for individual patients. For instance, OncoIMPACT [16] and DawnRank [17] prioritize patient-specific driver genes by exploiting the perturbations of the transcriptional programs through molecular networks. However, these approaches apply the same aggregated gene network to all patients, which may reduce the personalized information. Additionally, some approaches, such as SSN [18], SCS [19], and paired-SSN [20], have been proposed based on individual networks. The SSN algorithm constructs individual perturbation networks based on the expression data of diseased individual samples against a group of given control samples [18]. The importance of each edge was quantified using Pearson’s correlation coefficient to identify personalized driver genes. Similarly, SCS uses a random walk with a restart algorithm to construct personalized networks based on mutation data, expression data, and protein interaction network data [19]. The network-controlled strategy assesses the effect of mutations on expression patterns to identify personalized driver genes [19]. Paired-SSN uses paired sample expression data, i.e., normal and diseased data from the same sample, to construct individual networks [20]. It then identifies individual driver genes using a cybernetic approach, which provides a more personalized assessment of cancer driver genes [20]. Pham et al. proposed the pDriver [21] method, which constructs gene regulatory networks for each sample and uses network control strategies to identify personalized driver genes, including coding and non-coding genes. This method takes into account the use of known driver genes to localize to the patient’s personalized driver genes, which is not considered by most other methods [21]. PRODIGY [22] identifies driver genes by optimizing the cost of subtrees in the gene interaction network as a score for mutated genes. However, PRODIGY [22] did not use known driver genes to localize to the patient’s personalized driver genes. Overall, challenges remain in identifying the driver genes for each patient. In terms of methodology, some network-based methods will miss predictions for some patients. Many GCN-based methods are used to handle various bioinformatic tasks, but they still lack individual driver gene prediction. In terms of feature fusion, how to make better use of multi-omics features is also an issue that needs to be addressed. 

In this work, we present a novel approach for predicting personalized cancer driver genes for individual patients, called PDGCN. It uses a GCN model with a conditional random field (CRF) layer to more adaptively fuse multi-omics features over a network of individual attributes. Unlike some previous methods, PDGCN constructs the networks for each patient, and then a GCN model with a CRF layer is used to learn the feature representation of the nodes by combining the structural features of the network with the biological property features of the genes. PDGCN obtains data from each sample for training, thus avoiding the possibility that individual samples may be missing to predict outcomes. Finally, the driver genes are obtained from the results predicted by the model. To evaluate the performance of the proposed model, we applied it to two TCGA datasets and compared it with other similar methods. The experimental results demonstrate that our model outperformed the other methods in detecting personalized cancer drivers in individuals. 

## 2. Materials and Methods 

### 2.1. Datasets

For this study, we downloaded two cancer datasets, those of adrenocortical carcinoma (ACC) and kidney chromophobe (KICH), through The Cancer Genome Atlas (TCGA) [22] Xena platform data portal [23] (http://xena.ucsc.edu/, accessed on 1 July 2022). Both datasets contained somatic mutation, expression, methylation, and copy number variation data. We selected only patients that had all four types of data, resulting in 77 samples for ACC and 66 samples for KICH. The STRING dataset (v11.0) [24] was used to construct individual networks for each patient. To obtain a positive set, we selected known cancer driver genes from the Network of Cancer Genes (NCG 6.0) database [25], which is a manually managed repository that collects well-studied cancer genes from various sources, and the Cancer Gene Census (CGC) from the COSMIC database (v90) [26], which is a popular cancer gene dataset containing 719 well-established driver genes. In addition, we collected the list of cancer type-specific genes published by Bailey et al. [27]. The above three components above made up the positive set. The negative set was formed by recursively filtering out the positive set from the processed data. We conducted a series of experiments on the two different datasets (i.e., ACC and KICH). For ACC, we generated 5467 positive and 15,448 negative datasets; for KICH, we generated 5982 positive and 13,815 negative datasets.

### 2.2. PDGCN

In this work, we propose a new framework called PDGCN that is based on a GCN and can predict driver genes at the individual level. PDGCN consists of two main steps, as illustrated in Figure 1. The first step (Figure 1a) is the construction of a patient–gene interaction network based on the known PPI network for each individual sample. Here, the genes we used were DEG specific to the cancer and mutated genes for each individual. The second step involves using the GCN to learn the representation of each node in the network. To force the aggregated representation of the neighbor nodes, a CRF layer was added to the GCN model. Nodes are scored based on the representation learned by the GCN, and the driver genes are identified. Next, we describe the above two steps in detail.

#### 2.2.1. Construction of Personalized Networks 

In this study, we focused on coding genes; hence, the raw data sets from the TCGA were preprocessed by filtering out abnormal data, which mainly contained non-coding genes. To construct the individual-driven network, we used the processed data as follows. Firstly, the mutated genes of each patient were extracted from the mutation data. Next, the DEGs for each cancer type were selected by intersecting the results of the Anovar and Limma tools with an adjusted *p* < 0.05 and Log2 (FC) > 2 or Log2 (FC) < −2, respectively. This strategy can decrease the bias of a method. Then, the mutated genes and those DEG-specific to the individual were combined to create a subset of genes for each patient. These genes were then combined with the STRING dataset to obtain a sample-specific network, which we refer to as the sample–gene interaction network. Each node in this network was accompanied by a sample ID and a gene symbol, and a neighbor matrix A was constructed using this network.

In the sample–gene interaction network, the node’s attribute features can be classified into three main types: molecular features, system-level features (gene properties), and network structure features obtained through individual networks. Molecular features are extracted from somatic mutation, methylation, copy number, and expression data. For somatic mutation data, we used non-silent mutation data at the Multi-Center Mutation Calling in Multiple Cancers (MC3) gene level, where “1” indicates a mutation, and “0” indicates no mutation. Methylation data are represented by the beta values of the DNA methylation profiles measured experimentally, which are continuous variables between 0 and 1. The CNV data were processed using the GISTIC2 method, and the expression data were processed using log2-transformation. System-level features were obtained from sysSVM (from sysSVM method proposed by Nulsen et.al.) [28], which are the features of genes in global attributes. Here, the PPI network features were removed because the network we used was different from the sysSVM method. The features retained included 18-dimensional features, such as the length of the gene, the number of protein structural domains, the age of the gene, and the necessity of the gene. We matched these features to the genes of the individual samples to form complete system-level features. To obtain network structure features, the Node2vec [29], a model inherited from the random walk model in the DeepWalk (v1.0.2) [30] algorithm, was used to obtain features of different dimensions. To select the proper dimensions of the features, we used 10-, 20-, and 30-dimensional features for the subsequent experiments. Finally, all three types of features were stitched together to form the feature matrix X, which was then normalized to have values between 0 and 1.

#### 2.2.2. Graph Convolutional Network for Node Embedding 

A GCN is a multilayer graph convolutional neural network that aims to learn the node embedding by implementing convolutional operations on a graph. The basic concept of a GCN is to utilize the properties of neighboring nodes to improve the classification results. The topology of the graph is important, as the nodes can have a varying number of neighbors [31]. A GCN is a first-order local approximation of spectral graph convolutions that can handle first-order neighborhood information in each convolutional layer. Additionally, the multi-layer convolution permits the transfer of multi-order neighborhood information. From the adjacency matrix A and the feature matrix X, the simple propagation rules for each layer in a GCN can be defined as follows: (1)$$H^{\left(l+1\right)}=\sigma ({\tilde{D}}^{{-}_{2}^{1}}\tilde{A}{\tilde{D}}^{{-}_{2}^{1}}H^{\left(l\right)}W^{\left(l\right)})\;\;\;\;\;\;$$
where $\tilde{A}=A+I$, in which I is the unit matrix; ${\tilde{D}}_{ii}=\sum_{j}{\tilde{A}}_{ij}$ represents the degree matrix; W denotes the learnable weight matrix; and σ denotes a nonlinear function, such as the ReLU activation function. H is the feature of each layer, and for the input layer, H is X. The first layer receives X as input, so $H^{\left(0\right)}=X$. The added self-connection matrix $\tilde{A}$ helps to preserve the original node signals and incorporates them into the Laplace smoothing process. 

#### 2.2.3. CRF Layer for Embedding Update 

The existing approach of the GCN considers all neighbors equally, so it cannot retain the similarity information of similar nodes when learning the node embedding. We enhanced the representational learning of the nodes by adding a CRF layer to encourage similar nodes to keep similar hidden features. The CRF is a probabilistic graphical model proposed by Lafferty et al. [32]. Combining the features of the maximum entropy model and the hidden Markov model, it is an undirected graph model that is commonly used to label or analyze sequence information, such as natural language text or biological sequences. The loss function of the CRF layer is as shown in Equation (2).
(2)$$l_{CRF}=\sum_{i=1}l\left(H_{i}\right)\;\;\;\;\;\;\;$$
(3)$$\;\;l\left(H_{i}\right)=\alpha {\left\|H_{i}-Q_{i}\right\|}_{2}^{2}+\beta \sum_{j\in M_{i}}g_{ij}{\left\|H_{i}-H_{j}\right\|}_{2}^{2}$$
where $l\left(H_{i}\right)$ is the loss of the $H_{i}$ layer, and it can be calculated using Equation (3). $Q_{i}$ denotes the initial embedding of node i obtained from the GCN convolution layer, and $H_{i}$ denotes the updated embedding of node i in the CRF layer. In addition, $g_{ij}$ denotes the importance of neighboring node pairs. $M_{i}$ is the neighborhood of node i, and α and β are the balance factors of the two parts of Equation (2). 

Motivated by Long et al. [33], we used self-attention [34] to distinguish the contributions of neighboring node pairs. The use of self-attention makes it easier to update the losses. Formally, the attention $g_{ij}$ between nodes i and node j in Equation (3) is defined as follows.
(4)$$\alpha_{ij}=att(W_{t}H_{i},W_{t}H_{j})$$
(5)$$g_{ij}=softmax(\alpha_{ij})=\frac{exp(a_{ij})}{\sum_{x\in N_{i}}exp(a_{ix})}\;\;\;\;\;\;$$
where att() denotes a single-layer feedforward network to perform attention, and $W_{t}$ denotes the potential trainable matrix.

#### 2.2.4. Overall Loss and Optimization

Using the original binary cross-entropy loss function would result in a bias in data prediction toward the side with more samples, given the unbalanced nature of our positive and negative dataset. To address this issue, we attached weights of different multiples to the positive set. The modified loss function is as shown in Equation (6).
(6)$$l_{\theta }=-(pylog(h)+(1-y)log(1-h))$$
where p is the value of the different weights we added to the positive set, h is the output of the network after the sigmoid activation function, and y is the original node label (0 or 1). Finally, the overall loss $l_{Total}$ is defined as follows:(7)$$l_{Total}=l_{CRF}+l_{\theta }\;\;$$

The ADAM optimizer was used to train the GCN model.

## 3. Results

This section is divided into three parts. Firstly, we provide a brief description of our experimental setup. Then, we discuss the performance of the model and analyze it at both the population and individual levels. Lastly, we analyze the predicted rare mutant genes with novel biomarkers.

### 3.1. Experimental Setup

We implemented our model using Python 3.7 as the compiler. To achieve optimal model performance, we split all labeled genes, using 25% as the test set and 75% as the training set. Then, we used a grid search within a reasonable parameter range to optimize the hyperparameters, including learning rate, weight decay, dropout rate, and epoch, and selected the best performance hyperparameter combination as the final parameters by conducting five-fold cross-validation on the training set. For the Adam optimizer in the constructed individual gene network, we selected a learning rate of 0.001, weight decay of 0.005, and a dropout rate of 0.1 for 3000 epochs. We implemented the experimental code based on the open source machine learning framework TensorFlow. The experiments were conducted using the Windows 10 operating system, with an Intel^®^ Core^TM^ (Intel, Santa Clara, CA, USA) i5-8265U, 1.60 GHz CPU, GTX1060 graphics card, and 16 GB RAM.

### 3.2. Evaluation Metrics

To verify the effectiveness of the proposed method, the metrics of accuracy (ACC), area under the curve (AUC), and area under the precision–recall curve (AUPR) were considered. $ACC=\frac{\left(TP+TN\right)}{\left(TP+FN+FP+TN\right)}$, and the AUC is defined as the area under the ROC curve. The closer the AUC is to 1, the higher the correct rate. Additionally, the AUPR is defined as the area under the P-R curve, and the P-R curve is a graph consisting of precision (P) and recall (R), where $P=\frac{TP}{\left(TP+FP\right)}$ and $R=\frac{TP}{\left(TP+FN\right)}\;$.

### 3.3. Effect of Node2vec Dimensions

In this study, the network features extracted using Node2vec, which is a graph embedding method that can automatically extract the spatial features of graphs, were combined with biological attribute features. To explore the effects of different network features’ dimensions, we spliced three different dimensional features generated by Node2vec with the original features and performed the normalization operation on both sets of features. The model using the added 10-dimensional spatial features is denoted as GCN-CRF-Node10; the different results are shown in Table 1. The experiments demonstrated that combining the spatial features of graphs with biological attribute features can achieve a better performance. Although the best performance was not achieved in all metrics, we selected the relatively good features, with 20 dimensions added for subsequent experiments.

### 3.4. Ablation Experiments

The proposed method is based on a GCN with a CRF layer and network structure features. To evaluate their effects, ablation experiments were conducted. Three model variations are shown in Table 2: GCN is the traditional method with basic features, GCN-CRF is the GCN method with an added CRF layer, and GCN-CRF-Node20 represents the GCN-CRF model with an added network structure extracted using the Node2vec method. The results demonstrate that the GCN model with the CRF layer could better aggregate the information of similar nodes compared to using GCN alone. Additionally, the network structure features were useful and important in improving the performance.

### 3.5. Effects of Different Weights in Loss Function

The original GCN loss function is a binary cross-entropy loss function, and here, the original loss function could not make accurate judgments due to the imbalanced number of positive and negative sets. Based on the MODIG method proposed by Zhao et al. [35], different multiples of weights for the positive set in the best model we obtained in the previous step were applied, and the results are shown in Table 3. From the table, it can be seen that appropriate weights can further improve the performance of the model. Finally, for the ACC data, we used three times the positive set weights, while for the KICH data, we used two times the positive set weights.

### 3.6. Analysis at the Population Level

In this section, we compare the performance of our method with those of some existing methods at the population level since there is no factual benchmark for personalized driver gene identification. We selected some representative approaches in the field, including a frequency-based method, DriverMAPS [36], a network-based method, HotNet2 [13], and a machine learning-based method, sysSVM, which is also a method for identifying personalized cancer driver genes. The reason we selected the DriverMAPS and HotNet2 methods is that they have the best overall performances among 12 methods according to ref. [37]. 

To evaluate the performance of our method for the identification of driver genes, CGC was used as a basic benchmark. We selected the top 100 genes predicted by different methods and adopted three metrics, precision (P), recall (R), and the F1-score, to measure the performances of these methods. P represents the fraction of correctly predicted driver genes among predicted driver genes, while R represents the fraction of correctly predicted partial driver genes among the CGC driver genes. The F1-score, a combined metric of precision and recall, can effectively assess the ability of predicting cancer driver genes on the CGC database. It is calculated as $F1-score=2*\frac{P*R}{P+R}$. The results are shown in Figure 2, and it can be seen that our method outperformed the other methods in all three metrics for both datasets. This indicates the effectiveness of the method in identifying cancer driver genes at the population level.

### 3.7. Analysis at the Individual Level

The proposed method targets individuals. To validate the performance of the method at the individual level, we compared the proposed method with the individual-level sysSVM method [28]. Specifically, we analyzed each sample predicted by both methods, including 77 samples from ACC and 66 samples from KICH. The results of the ACC are shown in Figure 3 (for the results of KICH, see the Appendix A), which indicates the number of driver genes in each sample according to the CGC database for the top 50 and top 100 driver genes predicted by the two methods. It is worth noting that our method performed better than sysSVM on most samples from both datasets, with sysSVM performing better on six samples from ACC and three samples from KICH. 

### 3.8. Analysis of Identifying Rare Drivers

In this study, PDGCN could identify not only frequently mutated genes, but also rare driver genes, which are defined as those with a mutation frequency of <2% in the total patient cohort. We selected the identified rare genes in different samples and analyzed them according to previous studies. The results (see Appendix A) show that the identified rare genes play important roles in cancer.

### 3.9. Survival Analysis 

To verify whether the genes identified using this method can differentiate the prognostic risk of cancer patients, we validated the identified genes by conducting a survival analysis. The top 50 candidate driver genes for cancer predicted using the PDGCN method were subjected to survival analysis using the online tool GEPIA2 (Gene Expression Profiling Interactive Analysis, http://gepia2.cancer-pku.cn) [38]. Genes with logrank *p* < 0.05 were considered significant biomarker genes [39]. Moreover, among these significant biomarker genes, those not in the CGC dataset were considered novel candidate biomarker genes. Our method identified nine novel biomarker genes in the ACC data and seven novel biomarker genes in the KICH data. All novel biomarker genes identified were also mapped for survival analysis. The results of the ACC data are shown in Figure 4 (for the results of KICH, see Appendix A). Furthermore, to validate the role of these novel biomarkers, a literature analysis was also conducted, and the results are shown in the Appendix A.

From the results, it can be seen that these novel biomarkers can efficiently distinguish the longer survival group from the shorter survival group.

### 3.10. Enrichment Analysis

To analyze the associations among the identified driver genes at the population level, we used the DAVID (the Database for Annotation, Visualization and Integrated Discovery) [40] online tool to perform KEGG (Kyoto Encyclopedia of Genes and Genomes) pathway enrichment analysis and GO (Gene Ontology) function enrichment analysis. KEGG is a database that specifically stores information on gene pathways in different species, while GO function enrichment analysis mainly annotates gene products in terms of the biological processes involved (GO-BP), cellular components (GO-CC), and molecular functions (GO-MF). The results are shown in Figure 5. For the ACC data, the KEGG pathway analysis mainly revealed that these genes are significantly enriched in some important cancer-related pathways. Regarding the GO functional enrichment, the identified driver genes are significantly enriched in cell migration regulation and promoter regulation in terms of GO-BP. These processes can interact with *Ago1* and positively affect gene expression in cancer cells [41]. Regarding GO-CC, these genes are significantly enriched in the cytoplasm, cytosol, and cytoplasmic membrane. And specific hydrolase activity in ACC is positively correlated with cytoplasmic activity [42]. In terms of GO-MF, the analysis showed significant enrichment in the same protein binding sites and binding to kinase proteins. Corresponding inhibitors have been used as therapeutic agents for ACC [43]. For the KICH data, the KEGG results focused on some cancer-related pathways. In terms of the GO functional enrichment, driver genes were significantly enriched in some GO-BP terms; for example, the transformation of cellular proto-oncogenes into oncogenes leads to the over-activation of these signaling pathways, which in turn interact with the PI3K-Akt and Ras-ERK pathways to dysregulate cancer signaling and generate tumor cells [44]. Regarding GO-CC, the analysis showed significant enrichment in the cell membrane, nucleus, and, to a lesser extent, chromosomes. Related studies suggest that the deletion of DNA from chromosomes may be a unique feature of KICH [45]. In terms of GO-MF, a large number of proteins were enriched in protein binding, among which parvalbumin may be the KICH marker that distinguishes primary from metastatic tumors [46] (see Appendix A). 

## 4. Conclusions and Discussion

Although research on cancer driver gene identification algorithms has made some progress, there is still a need for further improvements in overall performance and to address certain problems. Cancer samples are highly heterogeneous from one another, and with the increasing emphasis on precision and individualized medicine, it is necessary to develop individual driver gene prediction methods on a population basis. Some previous methods are more dependent on the gene interaction network constructed and tend to ignore the similarity in characteristics between genes and typical known driver genes. Furthermore, some methods fail to achieve better fusion when integrating genomic, transcriptomic, and other multi-omics data and network data simultaneously. With the continuous development and application of machine learning, machine learning approaches have become increasingly successful in addressing various important biomedical problems. Machine learning plays an increasingly important role in developing models for predicting cancer driver genes.

Considering the limitations of previous driver gene identification methods and the advantages of machine learning, we propose a new machine learning approach, PDGCN, which applies a GCN to drive gene identification at the individual level. We constructed individual sample–gene networks in which each node was accompanied by information about each specific sample and gene. Additionally, the biological attribute features and spatial features of the genes were combined to constitute the enhanced features of the genes. We used a GCN with a CRF model to enhance the feature representations. We evaluated the performance of the method with different experiments and found that the method was more effective than other existing methods in identifying cancer driver genes at the population level. Furthermore, it was also able to identify novel biomarker genes, most of which have been confirmed in the literature to be associated with cancer. Personalized rare driver genes have also been detected and confirmed in the literature. Moreover, the enrichment analysis demonstrated that the predicted driver genes are significantly enriched in the GO terms and KEGG pathways. Overall, these findings suggest that our proposed method, PDGCN, can provide new insights into the molecular regulatory mechanisms during cancer development.

Although we constructed different networks based on different patients, the number of nodes in most patient gene networks did not significantly differ. This is because our approach was to construct personalized gene networks based on DEGs for specific cancers and mutated genes for individual samples. Moreover, we used only the STRING database to determine the interrelationship of patient genes, which may have resulted in less cancer type-specific information in the network. As a future research direction, we can consider specifying DEGs in different samples to make the specificity of the samples more apparent. Additionally, we can apply multiple gene interaction databases to enrich the relationships among genes and generate more possibilities. Good features are the key to improving model performance, and we will also consider feature selection techniques to optimize the way features are combined to achieve better results.

## Figures and Tables

**Figure 1 biology-13-00184-f001:**
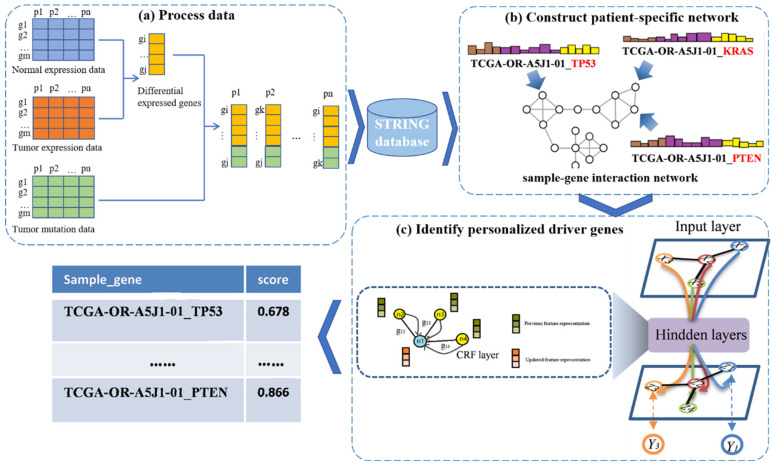
A flowchart of the PDGCN method. The method consists of three main steps. (**a**) Process data: A subset of genes for each sample are obtained using genes DEG specific to the cancer and mutated genes for each individual. (**b**) Construct a patient-specific network: A sample–gene interaction network for individual samples is constructed on the basis of the STRING database. (**c**) Identify personalized driver genes: Based on the representation of each node in the GCN, a CRF layer is inserted to force the aggregated representation of similar neighbors, and each node is scored based on the representation form learned by the GCN to identify driver genes.

**Figure 2 biology-13-00184-f002:**
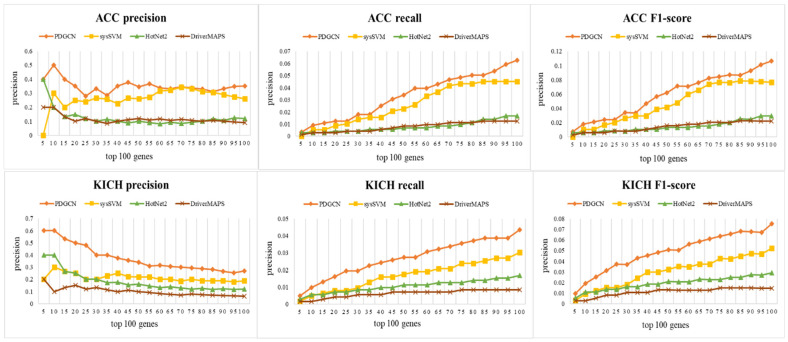
Performance comparisons (precision, recall, and F1-score) with PDGCN, sysSVM, HotNet2, and DriverMAPS methods on ACC and KICH datasets.

**Figure 3 biology-13-00184-f003:**
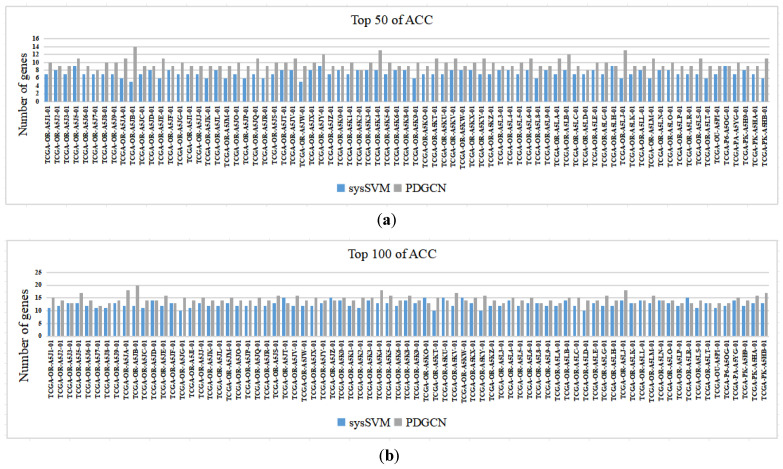
Comparisons of PDGCN and sysSVM at the individual level. The horizontal axis represents each sample, and the vertical axis represents the number of genes in the CGC: (**a**) Top 50 of ACC and (**b**) Top 100 of ACC.

**Figure 4 biology-13-00184-f004:**
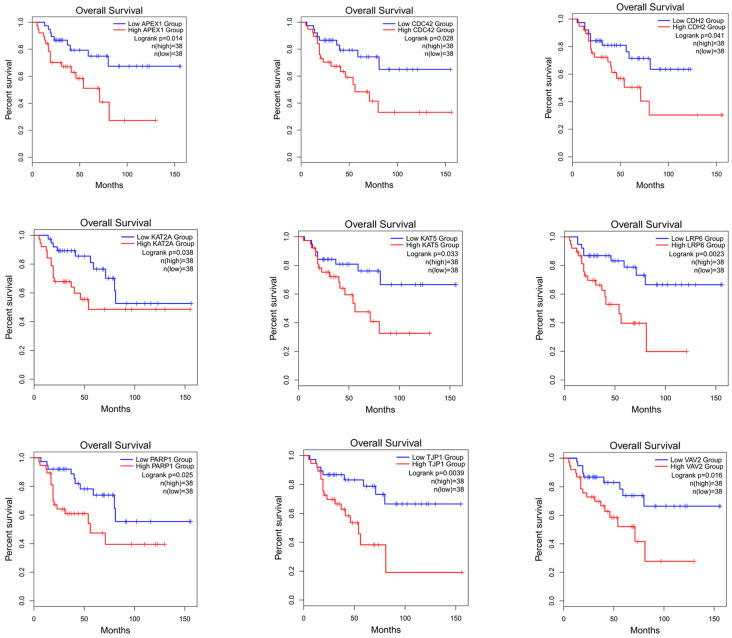
Survival analysis map of nine biomarker genes for ACC.

**Figure 5 biology-13-00184-f005:**
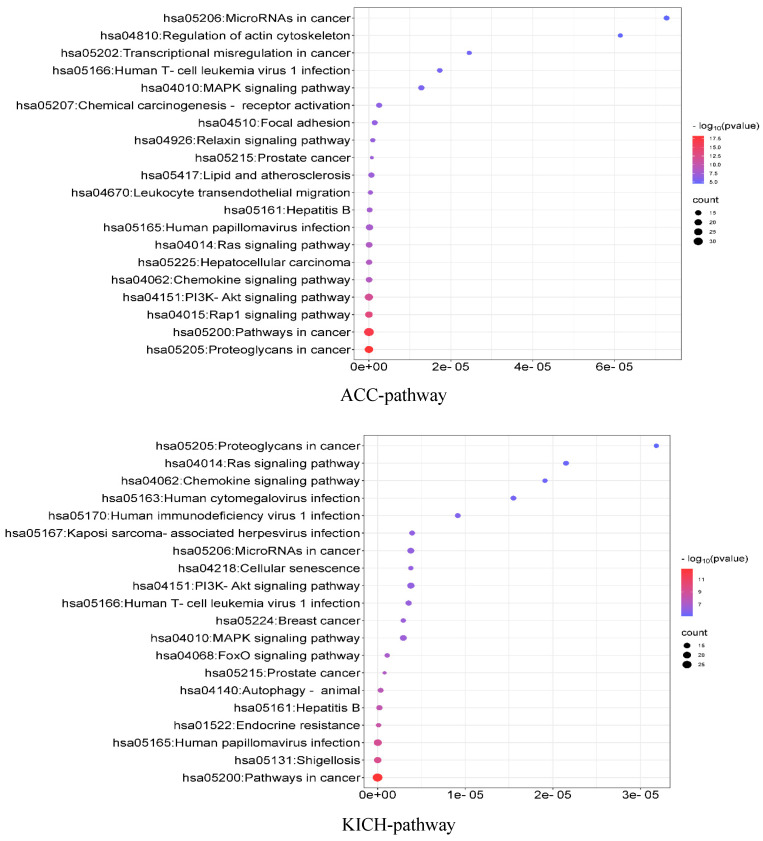
KEGG pathway enrichment analysis of ACC and KICH.

**Table 1 biology-13-00184-t001:** Results of different feature dimensions from node2vec. The bold font in the table indicates the best performances of the model under these conditions.

	ACC Data			KICH Data		
	Acc	Aupr	Auc	Acc	Aupr	Auc
GCN-CRF-Node10	0.849	**0.813**	0.890	0.865	0.810	0.912
GCN-CRF-Node20	**0.855**	0.802	**0.898**	0.858	**0.814**	**0.91** **9**
GCN-CRF-Node30	0.833	0.758	0.884	**0.866**	0.803	0.900

**Table 2 biology-13-00184-t002:** Results of ablation experiments. The bold font in the table indicates the best performances of the model under these conditions.

	ACC Data			KICH Data		
	Acc	Aupr	Auc	Acc	Aupr	Auc
GCN	0.821	0.772	0.879	0.804	0.807	0.904
GCN-CRF	0.850	0.796	0.892	0.821	0.813	0.902
GCN-CRF-Node20	**0.855**	**0.802**	**0.898**	**0.85** **8**	**0.814**	**0.91** **9**

**Table 3 biology-13-00184-t003:** Results of different weights for the positive set. The bold font in the table indicates the best performances of the model under these conditions.

	ACC Data			KICH Data		
	Acc	Aupr	Auc	Acc	Aupr	Auc
Weight of 1 time	0.855	0.802	0.898	**0.858**	0.814	0.919
Weight of 2 times	**0.861**	0.812	0.912	0.848	**0.816**	**0.929**
Weight of 3 times	0.851	**0.813**	**0.928**	0.830	0.803	0.916
Weight of 4 times	0.857	0.811	0.920	0.784	0.778	0.920
Weight of 5 times	0.846	0.790	0.920	0.767	0.795	0.897

## Data Availability

The somatic mutation, expression, methylation, and copy number variation data of ACC and KICH are publicly available on the Xena platform: http://xena.ucsc.edu/ (accessed on 1 July 2022). The string dataset is publicly available at https://cn.string-db.org/ (accessed on 1 July 2022).

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
