# Peer review of "Personalized Driver Gene Prediction Using Graph Convolutional Networks with Conditional Random Fields"

_biology, 2024, doi:10.3390/biology13030184_

Round 1

Reviewer 1 Report

Comments and Suggestions for Authors

Authors present PDGCN a deep learning approach, based on a graph neural network (GNN), in order to identify individual driver gene predictions. They use multi-omics features, such as transcriptomic, epigenomic and genomic data, along with network structure features obtained through Node2Vec algorithm. Then a graph convolutional network (GCN) is trained for driver gene mutation predictions. The model is improved by adding a conditional random field (CRF) layer, so that the representation learning of nodes is enhanced. 

Experimental tests are carried out considering two types of cancer, and the performances of PDGCN have been compared with other state-of-the-art predictors.

Experimental results show the better results obtained by PDGCN, furthermore, the top scored driver genes prediction have been analyzed using bioinformatics tools, such as enrichment tools.

The paper is scientifically clear and well written, even if English should need a full revision by some editing service. The proposed architecture is very interesting and the experiments, including the comparisons with other models, are fair.

I have some minor remarks:

  • Model parametrization should be discussed and the best hyperparameters should be reported.

  • Figure 2 is not clear. Please increase its size and explain the meaning of the x-axis

  • Please increase the size of Figure 3

  • Please increase the size of Figure 5, the pathway and GO terms can not be read

  • Please make available the source code of the model

Comments on the Quality of English Language

English should need a full revision by some editing service

Reviewer 2 Report

Comments and Suggestions for Authors

The paper is devoted to a development of new graph convolution networks for prediction of personalized driver genes. I expect the paper to be very interesting for readers, since the theme is rather hot for biomedecine. The results obtained by the authors look significant. However the presentation and design of the manuscript is awful and do not correspond to the high standards of the journal. Namely

1) there are a lot of unknown abbreviations like as MODIG, GEPIA2, DAVID, GO. With no any explanation and no any appropriate references. Of course, certain abbreviations can be used without details, but appropriate reference is required even for online tools. Moreover, I propose that the authors should provide a list of abbreviations in the supplementary information, since the number of the abbreviations is so large.

2) Figure 5 is completely unreadable, whereas figs 3 and 4 are only partly readable. I propose to place only part of them into the main text (say, fig.3 and b in the vertical position, fig.4 bottom line, fig.5 a and e, while other subplots are to be moved to the supplementary information.

3) Tables 1 and 2 look curious. What does mean bold fonts in the tables? Are the authors sure that 5 significant digits are required or the number of significant digits may be reduced in these tables?

5) Some reference sources are not found.

Comments on the Quality of English Language

I propose that the authors should provide a list of abbreviations in the supplementary information, since the number of the abbreviations is so large.

Reviewer 3 Report

Comments and Suggestions for Authors

Line 17: Give the PDGCN acronym as it is the first place you use it.

Line 35: Give ACC, KICH, TCGA expansions.

Line 50: “single nucleotide variations (SNVs)” should be written in capital letters instead. “Single Nucleotide Variations (SNVs)” Correct all abbreviations in this figure.

Make sure that there are no abbreviations that are not explained.

Line 137: Just give the link in the references. It will be sufficient to provide only numbered citations in the article.

Lines 151, 152: Error! There is a Reference Not Found text. Fix these errors.

Line 257, 258: How did you determine the epoch numbers and learning rate values?

Line 260: You do not need to provide a link.

Line 261: If you worked with a graphics card, please also write the model of the video card.

Line 281: Why is ACC in the first line dark? Is there any special reason?

Line 291: Why is ACC in the first line dark? Is there any special reason?

Line 297: Error in reference.

Line 303: Why is ACC in the first line dark? Is there any special reason?

Line 319: Error in reference.

Line 324: Graphs are given in Figure 2, but what do these graphs mean? An explanation must be given.

Line 330: Error in reference.

Figure 1, Figure 2, Figure 3 are not referenced in the text.

Line 392: The resolution of the texts on the Y axis of the graphics in Figure 5 should be increased. Some are unreadable.

The article is overall well planned. But there are shortcomings. There are figures that are not explained and figures that are not referenced in the text. The tables in which the results of the article are presented should also be explained one by one. The reasons for obtaining these results should be written. After these corrections are made, it should be checked again.
